# Synthesis, Characterization, Cytotoxicity, and Antibacterial Studies of *Persea americana* Mill. (Avocado) Seed Husk Mediated Hydronium Jarosite Nanoparticles

Nandipha L. Botha [1,2,*], Karen J. Cloete [1,2], Nolubabalo Matinise [1,2], Oladipupo M. David [3], Admire Dube [3] and Malik Maaza [1,2]

1  UNESCO-UNISA Africa Chair in Nanosciences & Nanotechnology Laboratories (U2AC2N), College of Graduate Studies, University of South Africa (UNISA), Muckleneuk Ridge, Pretoria 0003, South Africa; kaboutercloete@gmail.com (K.J.C.); matinn@unisa.ac.za (N.M.)
2  Nanosciences African Network (NANOAFNET), iThemba LABS-National Research Foundation, Somerset West 7129, South Africa
3  School of Pharmacy, University of the Western Cape, Bellville 7535, South Africa
*  Correspondence: bothanandipha9@gmail.com; Fax: +27-21-843-3525

**Abstract:** The application of nanotechnology in antimicrobial and cytotoxicity studies has recently been receiving increased interest. This paper report on the use of *Persea americana* Mill. (avocado) seed husk to synthesize hydronium jarosite nanoparticles in a facile, economical, and eco-friendly manner. We describe firstly the synthesis of hydronium jarosite nanoparticles using *P. americana* (avocado) seed husk aqueous extract as a reducing and chelating agent for the reduction of iron (II) sulfate heptahydrate. Secondly, we describe the characterization of the nanoproduct with scanning electron microscopy (SEM); energy dispersive X-ray spectroscopy (EDX); high-resolution transmission electron microscopy (HRTEM); X-ray powder diffraction (XRD) analysis; Fourier transform infrared spectroscopy (FT-IR); and, lastly, the cytotoxicity and antibacterial effect of hydronium jarosite nanoparticles using murine macrophage cells (Raw 264.7) cell lines, Gram-negative (*Escherichia coli*), Gram-positive (*Staphylococcus aureus*), and methicillin-resistant *Staphylococcus aureus*. These hydronium jarosite avocado seed husk-mediated nanoparticle-coated 2D sheets did not show any antibacterial activity against the bacteria tested but did show concentration-dependent cytotoxicity. Further research is required to optimize the antibacterial properties and reduce the cytotoxicity of this nanomaterial synthesized using green nanochemistry.

**Keywords:** avocado; *Persea americana*; phytosynthesis; nanoparticles; hydronium jarosite

## 1. Introduction

Jarosite, a secondary iron sulfate mineral is indirectly synthesized by oxidation of sulfide minerals. Minerals belonging to the jarosite series [1] comprise various isostructural compounds that generally occur in acidic environments such as mines, at the surface or near-surface conditions [2]. Their general formula is $A_{1-x}(H_3O)_x Fe^{3+}{}_{3-y}(SO_4)_2(OH)_{6-3y}(H_2O)_{3y}$, where the A could be (K, Na, $H_3O^+$, $NH_4$, Ag and Tl) or Pb cation [2]. One of the rare natural compounds in the jarosite minerals group contains hydronium ion [2,3]. Jarosite minerals have been synthesized using different methods that include chemical synthesis [4] and the biological oxidation of Fe [5], with acidity and temperature being shown to be significant factors affecting hydronium jarosite formation [6]. Unfortunately, phyto--engineering has not been explored enough in the synthesis of any jarosites, either as a mineral compound or at the nano-scale range. Phyto-engineering that employs plant materials to produce safe and biocompatible nanomaterials has received considerable interest in recent years due to the low cost, non-toxic, environmentally friendly, timeous, and 'one-pot' synthesis methodology employed during nanomaterial manufacturing [7]. Plant phytochemicals or

phyto-active including phenolic compounds have also been shown to serve as excellent reducing and chelating agents in the synthesis of smart nanomaterials harboring unique physico-chemical properties [8]. An example of a plant material that harbors a rich source of phytochemical compounds includes agro-waste material such as *Persea americana* Mill. (avocado) seeds [9]. When the avocado seed extract was used as a chelating and reducing agent for iron sulfate, hydronium jarosite sheets coated with nanoparticles were obtained. This study further elucidated the electrochemical and magnetic properties of the avocado seed-mediated hydronium jarosite nanomaterials [10].

The biomedical applications of hydronium jarosite nanomaterials synthesized using green methods have, however, not been investigated. We are also not aware of any studies on the synthesis of hydronium jarosite nanomaterials using specifically the husk of the avocado seed. Avocado seed husk is particularly rich in phytocompounds with high antioxidant potential which may potentially mediate the cytotoxicity of a nanomaterial when the seed husk is used in green nanochemistry [11]. Furthermore, since it is known that hydronium jarosites exhibit antimicrobial properties [12], hydronium jarosite nanoparticles may exhibit antimicrobial potential due to their unique physico-chemical makeup.

Here, we report on the synthesis of a rare type of mineral—hydronium jarosite—in the form of a nanomaterial by using *P. americana* seed husk aqueous extract as a reducing and chelating agent for the reduction of iron (II) sulfate heptahydrate. We further report on the structural characterization, cytotoxicity, and antimicrobial studies of the nanoproduct against murine macrophage cells (Raw 264.7) cell cultures and common infectious bacteria such as Gram-negative *Escherichia coli*, Gram-positive *(Staphylococcus aureus)*, and methicillin-resistant *Staphylococcus aureus* (MRSA). This study may have unique implications for the use of an agro-waste product—avocado seeds—as a sustainable source of nano-antibacterial.

## 2. Materials and Methods

Chemicals and materials used for nanoparticle synthesis: analytical grade iron (II) sulfate heptahydrate and alconox were purchased from Sigma-Aldrich. All aqueous solutions were prepared with double-distilled deionized (DI) $H_2O$. The glassware used in all experiments was washed with alconox and rinsed thrice with DI $H_2O$.

To prepare the aqueous avocado seed husk extract, the air-dried avocado seeds were collected from the waste. Seeds were separated from the husks and the seed husks were washed with deionized water and dried in air for 24 h. The dried seed husks were then crushed to powder using a blender; 10 g of the powder was added to 200 mL of deionized water and heated at 70 °C for 1 h and allowed to cool at room temperature to obtain the extract. The synthesis of hydronium jarosite nanoparticles was completed by following a nanochemistry approach and the method of Bhattacharjee et al., 2021 [13]. Next, 100 mL solution of $FeSO_4 \cdot 7H_2O$ was added to 100 mL of the extract and stirred for 3 h at 70 °C as the black precipitate formed. The aqueous mixtures containing avocado seed husk extract and nanoparticles were left to cool. For downstream analysis, the nanoparticles were separated from the solution and isolated using centrifugation at 4000 rpm for 20 min. The pellet obtained after centrifugation was washed with DI $H_2O$ and dried in a Memmert hot air oven at 50 °C for 2 h.

Characterization techniques: Morphological and elemental analysis was characterized using a Tescan MIRA3 SEM, whilst elemental composition was determined with EDX performed using a Nova NanoSEM at 20 kV. A Balzer's evaporation coater was used to carbon coat SEM samples. HRTEM images were captured using a Tecnai F20 FEG-TEM in bright field mode at 200 KeV, ImageJ was used to evaluate particle size distribution. Phase identification was done using XRD (Bruker Advanced AXS D8 diffractometer with monochromated Cu K$\alpha$ radiation [$\lambda$ (wavelength) = 1.54060 Å] operating in the Bragg–Brentano geometry and a step size of 0.2° with 0.2 s per step for angles between 5° and 85° was used for phase identification and sample crystallography. To characterize chemical bonds and functional groups, Fourier transform infrared spectroscopy (FTIR) using potas-

sium bromide pellet technology and a PerkinElmer 100 Spectrometer in the wave number 400–4000 cm$^{-1}$ range was used.

Cytotoxicity: The murine macrophage cells (Raw 264.7) was purchased from the American Type Culture Collection (ATCC TIB-71, Manassas, VA, USA). Standard tissue culture conditions were used to maintain the cells in a complete medium. Cells were sub-cultured approximately every 3–4 days at 37 °C in a humidified 5% $CO_2$ environment, using DMEM (Lonza, Cape Town, South Africa), 10% heat-inactivated fetal bovine serum (FBS), glutaMAX, antibiotic/antimycotic (Sigma-Aldrich, St. Louis, MO, USA), and gentamicin were added as supplements (Sigma-Aldrich, St. Louis, MO, USA).

To evaluate the cytotoxicity of hydronium jarosite (MTT) (3-(4,5-dimethylthiazol-2-yl)-2,5-diphenyltetrazolium bromide assays were employed as previously described with minor modifications. MTT assay is based on the ability of nicotinamide adenine dinucleotide phosphate (NADPH)-dependent cellular oxidoreductase enzymes to reduce the tetrazolium dye MTT to its insoluble formazan (purple color). Briefly, cells were seeded at $5 \times 10^4$ cells per well in a 96-well plate and incubated as earlier described using the cell culture standard procedure. After treatment with the NPs for 24 h, 48 h, and 72 h, the test medium was removed, and cells were incubated with 100 μL of MTT solution (0.5 mg/mL MTT diluted in the media without FBS) for 1 h at 37 °C, 5% $CO_2$. Subsequently, the MTT solution was removed and 100 μL of DMSO was added to each well. Optical density (OD) was read on a microplate reader at 550 nm, (FLUOstar Omega, BMG Labtech). Cell viability for each treatment was calculated as the ratio of the mean optical density of replicated wells relative to that of the control.

Nanoparticle preparation for cell culture: 2 mg/mL stock solution of hydronium jarosite was prepared using deionized water. The NPs were sonicated in (QSonica, LLC, Newtown, CT, USA. Misonixsonicators, XL-200 Series) for approximately 5 min. Nanoparticles were freshly prepared prior to each experiment.

Microbial assay: The agar well diffusion assay was used to investigate the antibacterial activity of the hydronium jarosite NPs. Colonies of each selected bacterial strain from a fresh 24 h stock plate were inoculated (by streaking) on Muller Hinton (MH) agar plates. The plates were dried at room temperature under sterile conditions followed by boring of 9 mm diameter wells in them. Different concentrations of hydronium jarosite NPs (1000 μg/mL, 500 μg/mL, and 250 μg/mL) and standard drug ciprofloxacin were added in the wells as positive control and distilled water as negative control. Plates were incubated at 37° C overnight for 24 h. After incubation, the antibacterial activities of the nanoparticles were assessed by the Zone of Inhibition (ZI) around the wells as the microbe growth inhibition. Bacterium strains were purchased from the American Type Culture Collection (ATCC, Manassas, VA, USA).

Micro-plate dilution assay: This assay was used to determine the minimal inhibitory concentrations (MIC) of hydronium jarosite NPs. Briefly, serial dilutions of the hydronium jarosite NPs were added to Mueller Hinton (MH) broth, followed by the addition of overnight incubated bacterial suspensions (adjusted to 0.5 McFarland standard), and incubated at 37 °C for 24 h. The MIC was recorded as the lowest concentration of the hydronium jarosite NPs that inhibit bacterial growth. Ciprofloxacin was used as a positive control.

## 3. Results

### 3.1. TEM and SAED and SEM and EDX

The structural properties of the material were investigated using transmission electron microscopy selected area electron diffraction (SAED) and scanning electron microscope. Energy-dispersive X-ray was used for elemental analysis. In Figure 1, images (a), (b), (c), and (d) are TEM images. Image a shows the spherical nanoparticles with 5.5 nm average diameter monodispersed on the 2D sheets clearly visible in image b. This is in agreement with the TEM obtained when the avocado seed was used instead of the seed husk [10]. Image c shows the lattice fringes with the lattice spacing of 0.24 nm indicating

the crystallinity of the material as confirmed by the SAED image d and XRD patterns in Figure 2. SEM in image (e) suggests the surface morphology of the material to be rough. EDX confirmed the presence of the jarosite and $H_3O$-jarosite seed EDX shows K and P as extra elements due to the presence of the extract.

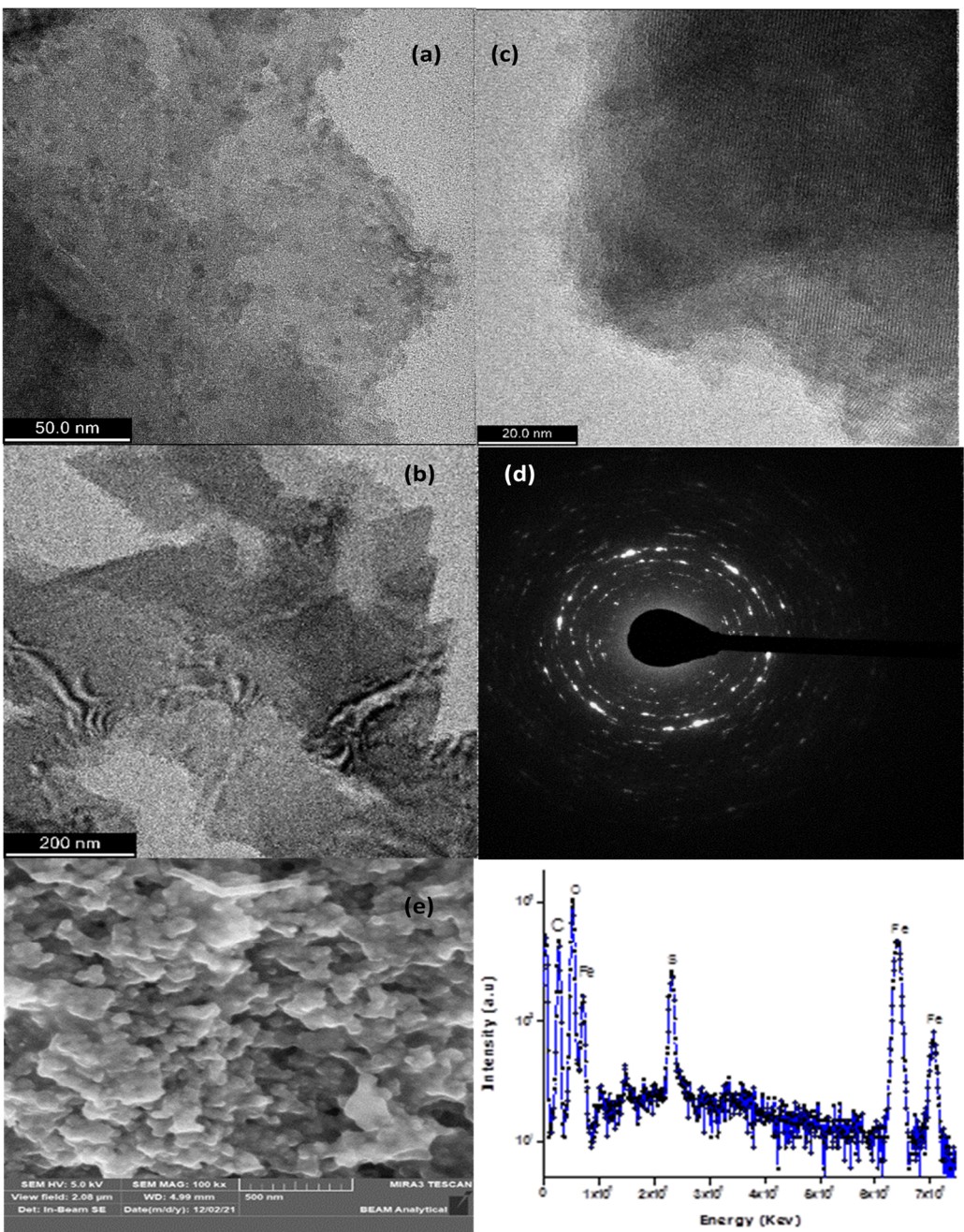

**Figure 1.** (**a–c**) TEM images showing the shape of the particles, the lattice fringes, and the triangular sheets, respectively. (**d**) SAED (**e**) SEM image displaying the surface morphology and EDX spectrum showing the elemental composition of the prepared hydronium jarosite husk.

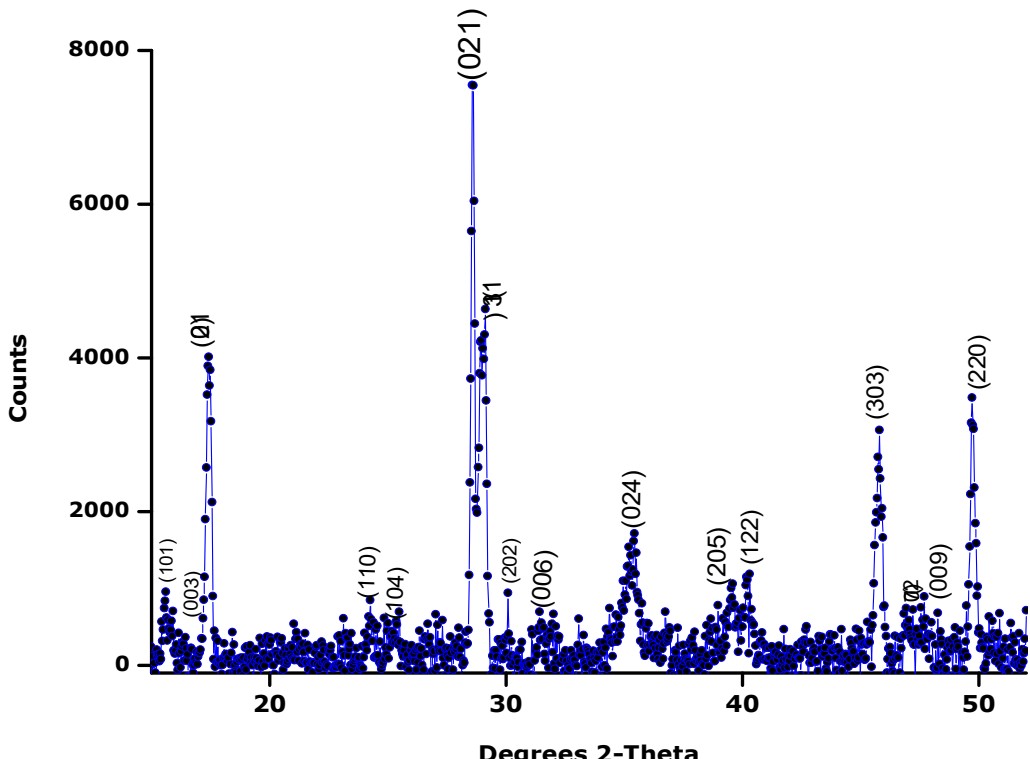

**Figure 2.** X-ray diffraction spectrum of the prepared hydronium jarosite husk showing the diffraction patterns.

### 3.2. X-ray Diffraction

The X-ray diffraction patterns of the green synthesized hydronium jarosite powder prepared using the *P. americana* (avocado) seed husk is shown in Figure 2. The observed peaks at 15.51°, 16.50, 17.44°, 24.31°, 25.38°, 28.58°, 29.12°, 330.10°, 31.34°, 35.34°, 39.6°, 40.23°, 45.75°, 47.17°, 48.33°, and 49.66° correspond to (101), (003), (012), (110), (104), (021), (113), (202), (006), (024), (122), (303), (027), (009), and (220), respectively. These peaks are in agreement with the rhombohedral hydronium jarosite phase (JCP2_31-0650). Sharp peaks show the crystallinity of the material. The obtained XRD results match the previously reported patterns of hydronium jarosite [10,14,15]. Furthermore, when we compare with the pattern of the hydronium jarosite we synthesized with the avocado seed extract, they show more crystallinity with the same phase [10].

### 3.3. FTIR

The comparison of the obtained FTIR results (Figure 3) also proves that the synthesized compound is a jarosite analog. As the spectra is showing the same stretching bands and vibrations as the various jarosite analogs reported [16]. The spectrum has a broad stretching band at 3370 cm$^{-1}$ is ascribed to $\nu$(O-H) vibration from the hydronium ions. The band around 1600, according to [2], is due to "overlapping bending modes of $H_2O$ and $H_3O^+$" which are mainly a result of the OH and $H_3O$ interaction forming water. The vibration at 1439 cm$^{-1}$ is attributed to $\nu$(O-H) and the doublets at 1097 cm$^{-1}$ and 1013 cm$^{-1}$ are assigned to $SO_4{}^{2-}$ vibrations. The intense band at 508 is assigned to the metal coordination band corresponding to $FeO_6$ coordination [6,17].

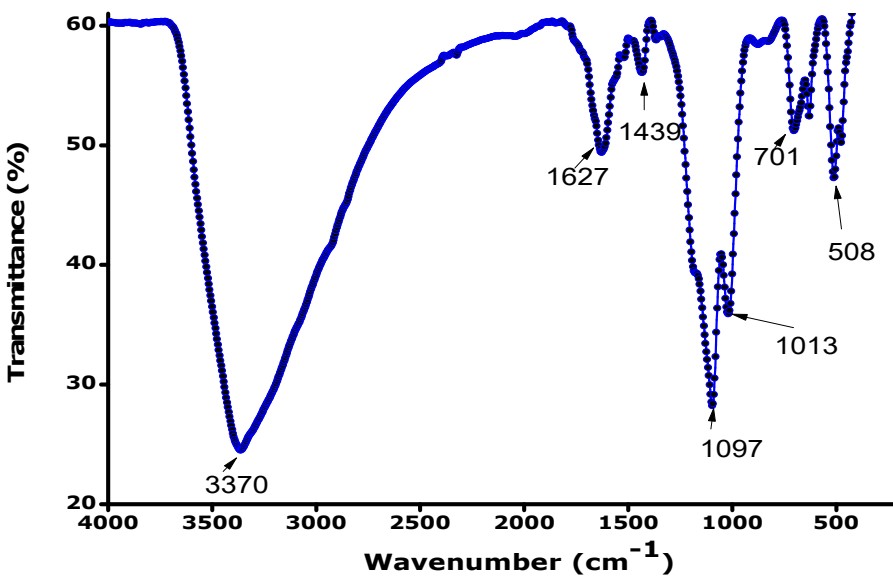

**Figure 3.** Fourier transform infrared spectrum of the prepared hydronium jarosite husk showing the vibrations of the functional groups present in the obtained material.

### 3.4. The Cytotoxicity and Antibacterial Effects

To evaluate the toxicity and antibacterial effect of hydronium jarosite nanoparticles using murine macrophage cells (Raw 264.7) cell cultures, *E. coli*, *S. aureus*, and MRSA. The result shows a concentration-dependent cytotoxicity from $\geq$100 μg/mL at 24 h, and $\geq$50 μg/mL for 48 and 72 h (Figure 4). Both the Gram-negative *E. coli* and Gram-positive *S. aureus*, as well as the MRSA bacteria tested were not susceptible to the hydronium jarosite NPs at the highest concentration of $\leq$1000 μg/mL (Figure 5).

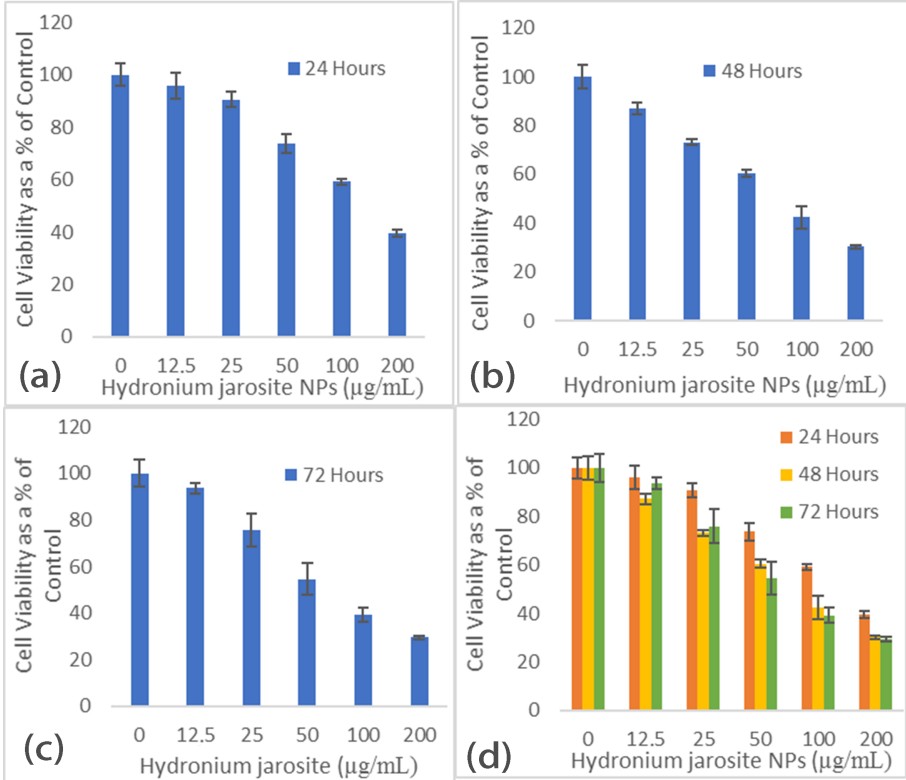

**Figure 4.** Shows the cell viability after (**a**) 24 h, (**b**) 48 h, (**c**) 72 h, and (**d**) cell viability after 24 h, 48 h, and 72 h exposure to different hydronium jarosite NPs concentrations; data represented as mean ± SD.

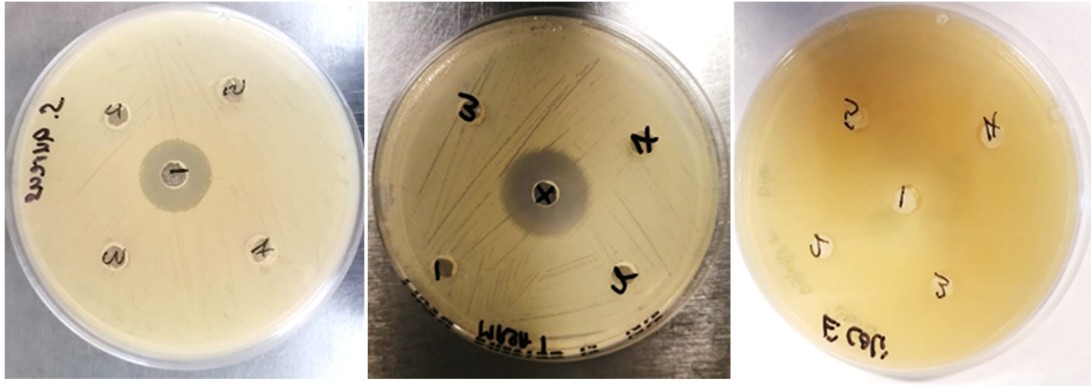

**Figure 5.** X—standard/positive control ciprofloxacin 10 µg/mL; 1—negative control (distilled water); 2—hydronium jarosite NPs (1000 µg/mL); 3—hydronium jarosite NPs (500 µg/mL); and 4—hydronium jarosite NPs (250 µg/mL) for methicillin-resistant *Staphylococcus aureus*; while 1—standard/positive control ciprofloxacin 10 µg/mL; 2—negative control (distilled water); 3—hydronium jarosite NPs (1000 µg/mL); 4—hydronium jarosite NPs (500 µg/mL); and 5—hydronium jarosite NPs (250 µg/mL) for *Staphylococcus aureus* and *E. coli*.

Micro-Plate Dilution Assay

In Figure 6 the 96-well plate (a) shows the hydronium jarosite NPs with *S. aureus* from a concentration of 1000 µg/mL A1–A6, 500 µg/mL B1–B6, 250 µg/mL C1–C6, 125 µg/mL D1–D6, 62.5 µg/mL E1–E6, 31.25 µg/mL F1–F6, 15.625 µg/mL G1–G6, and 78.13 µg/mL H1–H6. Controls are broth only 7A–7H, bacterial only (− control) 8A–8H, bacterial + antibiotic ciprofloxacin (+ control) 9A–9H, and the NPs only 10A–10H. The purple coloration shows the presence of bacteria while the blue shows the absence of bacteria. Plate (a) shows the hydronium jarosite NPs with the bacterial MRSA from a concentration of 1000 µg/mL A1–A6, 500 µg/mL B1–B6, 250 µg/mL C1–C6, 125 µg/mL D1–D6, 62.5 µg/mL E1–E6, 31.25 µg/mL F1–F6, 15.625 µg/mL G1–G6, and 7.813 µg/mL H1–H6. Controls are broth only 7A–7H, bacterial only (− control) 8A–8H, bacterial + antibiotic ciprofloxacin (+ control) 9A–9H, and the NPs only 10A–10H. The purple coloration shows the presence of bacteria while the blue shows the absence of bacteria. Plate (c) shows the hydronium jarosite NPs with *E. coli* from a concentration of 1000 µg/mL A1–A6, 500 µg/mL B1–B6, 250 µg/mL C1–C6, 125 µg/mL D1–D6, 62.5 µg/mL E1–E6, 31.25 µg/mL F1–F6, 15.625 µg/mL G1–G6, and 7.813 µg/mL H1–H6. Controls are broth only c7A–7H, bacterial only (− control) 8A–8H, bacterial + antibiotic iprofloxacin (+ control) 9A–9H and the NPs only 10A–10H. The purple coloration shows the presence of bacteria while the blue shows the absence of bacteria. These results corroborated the agar well diffusion assay and revealed that the bacteria tested were not susceptible to the hydronium jarosite NPs at the highest concentration of ≤1000 µg/mL.

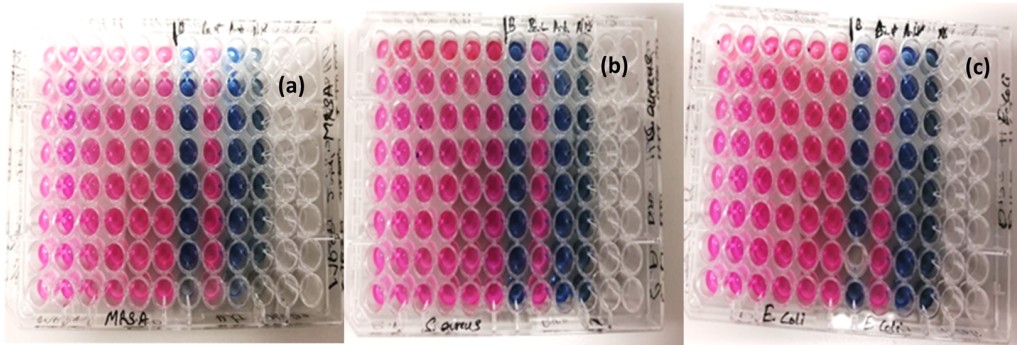

**Figure 6.** Micro-plate dilution essays, (**a**) methilin-resistant *Staphylococcus aureus*, (**b**) *Staphylococcus aureus*, and (**c**) *Escherichia coli*.

## 4. Discussion

The plant extracts are known for their various phytocomplexes, some of which have been proven to have the ability to reduce metal salts to metal nanoparticles [18]. For instance, for metal nanoparticles, mostly silver and gold, it has been discovered that terpenoids [19], protein [20], flavonoid [21], functional groups like carbonyl, amino, and sulfhydryl [22], and polyols [23] from different plants play a role in gold nanoparticle surface capping and reduction of both gold and silver ions. Metal nanoparticles have been synthesized using plant extracts such as rosemary [24], olives [25], and green tea to name a few, and have been found to have antibacterial activity [26]. In this study, *P. americana* Mill. seed husk extract was used. The seed on its own contains compounds such as flavonoids, tannins, saponins, phenolic, and alkaloids which are known to be present in most plants [27]. We were specifically interested in whether hydronium jarosite nanoparticles can be reproduced using only the seed husk instead of the avocado seed and its potential application as an antibacterial agent.

Metal-based nanoparticles are currently being investigated for different applications. Among these metal-based nanoparticles, iron oxides have been synthesized and have been shown to possess antimicrobial activity [28] when they were tested against *E. coli*, *S. aureus*, *Pseudomonas*, and *Klebsiella pneumonia*. It was found that particle size and concentration had an effect [29]. Jarosite $((H_3O)Fe_3 + 3(SO_4)_2(OH))$ has been found to possess significant antibacterial properties. A study conducted by Sharma et al. (2018) demonstrated that jarosite nanoparticles exhibited a dose-dependent antibacterial effect against two bacterial strains, namely *E. coli* and *S. aureus*. The study further revealed that jarosite nanoparticles caused cell membrane damage, which ultimately resulted in bacterial death. These findings suggest that jarosite has potential applications as an antibacterial agent in various fields, including medicine and agriculture. However, the present study did not show any effect on the bacteria tested [12]. The micro-plate dilution essay result corroborated the agar well diffusion assay and revealed that the *E. coli*, *S. aureus*, and MRSA tested were not susceptible to the hydronium jarosite NPs at the highest concentration ≤1000 μg/mL. This clearly shows that the material has no antimicrobial activity.

Several studies have investigated the cytotoxicity of jarosite on various cell types, including human cell lines and aquatic organisms. One study conducted by Guo et al. (2016) evaluated the cytotoxicity of jarosite on human hepatoma HepG2 cells. The results showed that jarosite exposure induced dose-dependent cytotoxicity, leading to increased levels of reactive oxygen species and apoptosis in the cells. The study suggested that jarosite exposure may cause oxidative stress and apoptosis in human hepatoma cells [30]. Another study by Zhang et al. (2019) investigated the cytotoxicity of jarosite on the aquatic organism *Daphnia magna*. The results showed that jarosite exposure resulted in decreased survival and reproduction of the organisms, as well as increased levels of oxidative stress and DNA damage. Overall, these studies suggest that jarosite can have cytotoxic effects on various cell types and organisms. In this study, we used hydronium jarosite which showed concentration-dependent cytotoxicity from ≥100 μg/mL at 24 h and ≥50 μg/mL for 48 and 72 h. On the other hand, the cytotoxicity effect observed for hydronium jarosite nanomaterials can support its potential application as an anticancer agent [31]. Further studies are needed to fully understand the mechanisms of jarosite toxicity and the potential risks it may pose to human health and the environment [32].

## 5. Conclusions

In this study, hydronium jarosite material was successfully synthesized via the green process where the avocado seed husk extract was used to reduce the iron sulfate salt. Particles corresponding to the rhombohedral phase of hydronium jarosite were obtained. TEM showed 2D sheets coated with spherical nanoparticles of 5.5 nm average diameter Even though it is known that plants contain phytocomplexes, for further studies it is vital to investigate and identify the compounds responsible for the reduction of iron(II) sulfate heptahydrate salt into the nanoparticle coated 2D sheets. These nanoparticles-coated 2D

sheets did not show any antibacterial activity against the bacteria tested but did show concentration-dependent cytotoxicity. Although a green approach was used for synthesis, the nanomaterial still induced cytotoxicity. This study is the beginning of further research on the green meditated hydronium jarosite, therefore no conclusions can be drawn on their ability as an alternative to existing therapeutics. Further studies are needed to elucidate the mechanism of hydronium jarosite nanoparticle-induced cytotoxicity combined with optimization of its antimicrobial properties.

**Author Contributions:** Conceptualization, M.M., N.L.B. and K.J.C.; lab work, N.L.B.; writing—original draft preparation, N.L.B.; writing—review and editing, K.J.C.; plant extraction, N.M.; the cytotoxicity and antibacterial effects, A.D. and O.M.D.; supervision, M.M. All authors have read and agreed to the published version of the manuscript.

**Funding:** This research received no external funding.

**Data Availability Statement:** All data is available upon request from the corresponding author.

**Acknowledgments:** The authors would like to show gratitude to iThemba labs (NRF), UNISA, and The School of Pharmacy in UWC.

**Conflicts of Interest:** The authors declare no conflict of interest.

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
