# Peer review of "Synthesis, Characterization, Cytotoxicity, and Antibacterial Studies of Persea americana Mill. (Avocado) Seed Husk Mediated Hydronium Jarosite Nanoparticles"

_applsci, doi:10.3390/app13158963_

Round 1
Reviewer 1 Report
All manuscript must be checked have a several type mistake (see enclosed corrected)
Change uppercase to lowercase
Normal to italics

Author Response
Thank you for your comments.
All uppercases were changed to small letters
All normal were changed to italic letters
The manuscript was reread and improved.
Reviewer 2 Report
This manuscript deals with "Phytosynthesis, characterization and antimicrobial studies of Persea americana (avocado) seed husk mediated hydronium jarosite nanoparticles" I suggest a minor correction and require a detailed clarification. A correction should be addressed by the authors as follows: The abstract is not well organized; the sentences are incomplete, and there is no sense of continuity. It would be feasible if you included the significance of the current study in the abstract. A brief description of how the authors selected information from the literature in the databases, as well as what time period they searched for, is missing. The authors should justify and expand the information on the advantages of Persea americana (avocado) seed husk mediated hydronium jarosite nanoparticles for biomedical applications. Authors should specify the main experimental conditions used based on the evidence from the literature. Where they briefly describe the most important data reported in the literature in a homogeneous manner and reinforce the relevance of Persea americana (avocado) seed husk mediated hydronium jarosite nanoparticles as novel alternatives. Authors should discuss whether the use of Persea americana (avocado) seed husk mediated hydronium jarosite nanoparticles represents a solid alternative to existing therapeutics. Also, please discuss the use of method using green nanomaterials to targeting cells and mitochondria . Please add the below studies to your manuscript in the discussion section and bold your study novelties:
-Ramazanli, V. N., & Ahmadov, I. S. (2022). SYNTHESIS OF SILVER NANOPARTICLES BY USING EXTRACT OF OLIVE LEAVES. Advances in Biology & Earth Sciences Vol.7, No.3, 2022, pp.238-244 -
-Baran, A., Fırat Baran, M., Keskin, C., HatipoÄŸlu, A., Yavuz, Ö., Ä°rtegün Kandemir, S., ... & Eftekhari, A. (2022). Investigation of antimicrobial and cytotoxic properties and specification of silver nanoparticles (AgNPs) derived from Cicer arietinum L. green leaf extract. Frontiers in Bioengineering and Biotechnology, 10, 263.
Author Response
- The abstract was rearranged and improved.
- This piece of work is not a systematic literature review and hence does not require a methodological description of the literature search.
- The advantages of using avocado seed husk is highlighted in the introduction line 58. However, we tried not to dwell much on the advantaged of using plant extract as a lot of reviews have been published on that subject.
- The experimental section was updated and all conditions and durations are now specified.
- It is in the early stages of this research to conclude that this material can replace the existing therapeutics, we highlighted this in the conclusion.
- Our focus is the synthesis of the nanomaterials using plant extract we did no go as far as studying the mechanism of the biological application of the materials
- The two references have been included
Reviewer 3 Report
Botha et al submitted a manuscript titled "Phytosynthesis, characterization and antimicrobial studies of Persea americana (avocado) seed husk mediated hydronium jarosite nanoparticles" for publication in MDPI Applied Sciences.
Though the work and design are interesting, but the writing is rudimentary. The paper has not been written to an acceptable extent for publication.
Rethink the title. It is not clear what you want to convey through title. It should be both comprehensive and succinct.
The figure descriptions need to be much more elaborate. Refer other celebrated publications in the field. Figure descriptions should include one statement of results, brief description of methods used to generate this piece of information in that figure, and detailed description of every component of the figure, including footnotes in graphical representations. Include significance values in all graphs and mention the name the tests used in descriptions.
All scientific names of plants and micro organisms need to be italicized.
Remove hyphens in unnecessary places (eg. in line 18)
Remove capitals in unnecessary places (eg. seed husk in line 75).
Author Response
Thank you for the comments.
1. We have changed the tittle.
2. The figure descriptions were revised
3. All scientific names of plants and micro organisms are italicized.
4. Unnecessary hyphens are removed
5. All capital letters changed to small letters
Reviewer 4 Report
Dear authors,
The manuscript entitled „Phytosynthesis, characterization and antimicrobial studies of Persea americana (avocado) seed husk mediated hydronium jarosite nanoparticles” Botha et al. describes a report on the use of Persea americana (avocado) seed husk to phytosynthezise hydronium jarosite nanoparticles in a facile, economic, and eco-friendly manner.
First, the authors must carefully read the Instructions for authors.
Please write in italics the name of the plant in Lines 16, 48, and 52 and the name of the microorganisms in Lines 23-24, 198-199, and 221-222.
Figure 1. Photos are of poor quality.
In the section Materials and Methods, it is not described how the tests were carried out with microorganisms.
In 3.4.2. Micro-plate dilution assay section, the letters of the plates must be the same a=a, A=A in text and in pictures.
Author Response
Thank you for you comments.
- All plant names and the name of the microorganisms are italicized
- Figure 1 removed as we do not have high quality photos however, we made sure that everything is explained in the text
- the numbering in the imaged is now the same as in the text
Reviewer 5 Report
Dear authors, After reviewing the following manuscript entitled "Phytosynthesis, characterization and antimicrobial studies of Persea americana (avocado) seed husk mediated hydronium jarosite nanoparticles" and with reference number (Applied Sciences - 2329550), I sent the following comments and observations that the authors should attend to before its publication in this journal. I appreciate the work of the authors, but please resolve the following data: The title should be changed. Considering the research carried out, I believe that phytosynthesis was not done, but only the incorporation of some biocompounds into nanostructures. Better to use the term synthesis or obtaining. If determinations were also made of antitumor activity, then why is reference only made to antimicrobial activity in the title? In the title but also in the manuscript, the binomial name of the studied plant should be written correctly, respectively Persea americana Mill.In the introduction, it would be good to specify more precisely the results obtained when determining the biological activities pursued in the study.
of the conducted research.
I recommend that the information in the article be reevaluated by the team of researchers.
Author Response
Dear Reviewer thank you for the comments.
- We have changed the title and we have also included the cytotoxicity in the title, we also corrected the name of the pant on the title and also in the text
- The experimental is improved the conditions and more detail is provided
- Units "ml" is changed to "mL"
- All footnotes of the figures have been modified
- It is known that phytocomplexes are extracted from plants, in this study it is not known which group of bioactive compounds were tracked.----This is not the focus of this study and previous studies have already screened the phytocompounds of avo seeds...
Round 2
Reviewer 1 Report
The actions were ok
Reviewer 4 Report
Dear authors,
The manuscript entitled „Phytosynthesis, characterization and antimicrobial studies of Persea americana (avocado) seed husk mediated hydronium jarosite nanoparticles ” Botha et al. describes a report on the use of Persea americana (avocado) seed husk to phytosynthezise hydronium jarosite nanoparticles in a facile, economic, and eco-friendly manner.
After reading the manuscript, I did not notice any significant errors, nor spelling and stylistic errors, English language and style are also fine. Conclusions are adequate for the conducted research. For my part, I would recommend accepting this manuscript in its present form.
Reviewer 5 Report
I agree with the publication of the article.